# Psychometric properties of the Patient Reported Outcomes, Burdens and Experiences (PROBE) questionnaire

Chatree Chai-Adisaksopha,[1,2] Mark W Skinner,[2,3] Randall Curtis,[4] Neil Frick,[5] Michael B Nichol,[6] Declan Noone,[7] Brian O'Mahony,[7,8] David Page,[9] Jeffrey Stonebraker,[10] Lehana Thabane,[2,11] Mark Crowther,[1,2] Alfonso Iorio[1,2]

For numbered affiliations see end of article.

**Correspondence to**
Mr Mark W Skinner;
mskinnerdc@gmail.com

## ABSTRACT

**Objective** To assess the psychometric properties of the Patient Reported Outcomes, Burdens and Experiences (PROBE) questionnaire.

**Methods** This study was a cross-sectional, multinational study. Participants were enrolled if they were more than 10 years old and people with haemophilia A or B or people without a bleeding disorder. Participants were invited through non-governmental patient organisations in 21 countries between 01/27/2016 and 02/23/2017. The following psychometric properties: missing data, floor and ceiling effects, exploratory factor analysis and internal consistency reliability were examined. A PROBE Score was derived and assessed for its convergent and known groups validity.

**Results** The study analysed the data on 916 participants with median age of 37.0 (IQR 27.0 to 48.0) years, 74.8% male. In the domain assessing patient-reported outcomes (PROs), more than 15% of participants presented a ceiling effect for all items but two, and a floor effect for one item. Factor analysis identified three factors explaining the majority of the variance. Cronbach's alpha coefficient indicated good internal consistency reliability (0.84). PROBE items showed moderate to strong correlations with corresponding EuroQol five dimension 5-level instrument (EQ-5D-5L) domains. The PROBE Score has a strong correlation ($r$=0.67) with EQ-5D-5L utility index score. The PROBE Score has a known groups validity among various groups.

**Conclusions** The results of this study suggest that PROBE is a valid questionnaire for evaluating PROs in people with haemophilia as well as control population. The known-group property of PROBE will allow its use in future clinical trials, longitudinal studies, health technology assessment studies, routine clinical care or registries. Additional studies are needed to test responsiveness and sensitivity to change.

**Trial registration number** NCT02439710; Results.

## BACKGROUND

Haemophilia is an inherited X linked recessive bleeding disorder characterised by the reduction or absence of blood coagulation factor (F) VIII (haemophilia A) or FIX (haemophilia B). Severity of haemophilia is

### Strengths and limitations of this study

► The Patient Reported Outcomes, Burdens and Experiences (PROBE) questionnaire was conducted to assess patient-reported outcomes in people with haemophilia (PWH). This tool assesses domains pertaining to general health status, haemophilia-related health status and health-related quality of life.
► The psychometric analyses demonstrate the validity and internal consistency of the PROBE questionnaire.
► This study was conducted in a large sample of PWH and participants without bleeding disorders from multiple countries.
► The responsiveness of the measurement was not investigated in this current study.

categorised by the baseline factor level (mild; factor level >0.05 to <0.40 IU/mL, moderate; factor level 0.01–0.05 IU/mL and severe; factor level <0.01 IU/mL).[1] Coagulation deficiency renders patients prone to abnormal bleeding. Symptoms of haemophilia vary depending on the severity of haemophilia, mechanism and severity of injury and affected organs. People with haemophilia (PWH) commonly present with haemarthrosis, gastrointestinal or genitourinary tract bleeding, intramuscular bleeding or intracranial bleeding.[2–6]

Life expectancy of PWH substantially improved with factor replacement therapy.[7] However, PWH who live longer encounter more chronic complications from both haemophilia-related conditions and degenerative diseases that occur in normal population. Chronic degenerative joint diseases are found in 90% of PWH by the second or third decade of life.[8] PWH with recurrent joint bleeding suffer from chronic pain, limitation of range of motion and disability.[9] HIV and hepatitis C virus (HCV) infections are prevalent among PWH prior to the implementation of intensive viral screening in plasma-derived factor

BMJ

concentrates and the use of recombinant factor concentrates.[10] One of the major consequences of chronic HCV infection is cirrhosis, resulting in end-stage liver disease which is the most common cause of death in PWH.[10] Moreover, 43% of cancers diagnosed in PWH were related to HCV infection.[11] Aged PWH are also affected by cardiovascular diseases. A retrospective study using an administrative database of 3422 males with haemophilia reported a prevalence of ischaemic heart disease of 15% in PWH older than 60 years.[12] Risk factors of cardiovascular disease in PWH are equivalent to patients without haemophilia.[13] These long-term complications of haemophilia directly impact on health-related quality of life (HRQoL) in PWH.[14]

Patient-reported outcomes (PROs) are defined as any reports of status of patients' health conditions that come directly from the patients without interpretation by clinicians or anyone else.[15] PROs provide data obtained from patients including symptoms, frequency of symptoms, severity of symptoms, impact of disease on daily life, disability and perfection of patients towards diseases and treatments.[16] Thus, PROs have been increasingly valued by researchers, stakeholders, policy makers and health technology assessment agencies.[17–20] Recently, the International Society for Pharmacoeconomic and Outcomes Research (ISPOR) Clinical Outcome Assessment Emerging Good Practices Task Force published the PRO and observer-reported outcome assessment in rare disease clinical trials.[21] This report demonstrated the challenges of assessing PROs in rare diseases, for instance, heterogeneity of disease severity and patient experience or understanding treatment benefit from the patients' perspective. Haemophilia, which is a rare bleeding disorder, exhibits various disease severity. Moreover, patients' perspective on their symptoms may be dissimilarly influenced by age, comorbid disease, inhibitor status, current treatment or progression of symptoms. Therefore, a haemophilia-specific PRO measure is essential for assessing outcomes in this patient population.

The Patient Reported Outcomes, Burdens and Experiences (PROBE) Project is a patient-lead research initiative. The main objectives of the PROBE Project are to develop a standardised PRO questionnaire and to develop a dedicated research network to generate and continuously update PROBE reference data. The feasibility study of the PROBE questionnaire was conducted in collaborations with non-governmental haemophilia patient organisations (NGOs) in 21 countries. Previously reported results demonstrated that the burden of the PROBE questionnaire implementation was minimal and the time required to complete the questionnaire was less than 15 min for most (71.3%) participants.[22] The objective of the current study is to assess the psychometric properties of the PROBE questionnaire.

## METHODS

### Patient and public involvement

The PROBE Project was initiated and led by investigators who are patients with haemophilia. Subsequently, the investigators identified and invited a group of national haemophilia patient organisations to participate in the PROBE Project to form a research network. The patient-important outcomes and metrics incorporated into the PROBE questionnaire were identified, developed and refined by the PROBE investigators and patient representatives from the participating national patient organisations (see acknowledgments). The patient organisations were then asked to enrol participants. Data from the PROBE study are analysed, summarised and disseminated to each patient organisation. Full development details of the PROBE questionnaire and patient-led research network are reported elsewhere.[23]

### Participant enrolment and study procedure

This study was designed as a cross-sectional assessment. Participants were enrolled through NGOs from 1/27/2016 to 2/23/2017. Participants were recruited if they were more than 10 years old and they were either PWH (haemophilia A or haemophilia B) or controls (participants without bleeding disorders). Participants were instructed to complete the questionnaire only once and answering for themselves, and parents or caregivers were instructed not to answer for their child. Although collected as part of the study, participants who identified themselves as carriers of haemophilia were excluded from the analysis. Patients with other bleeding disorders or an unknown bleeding disorder were also excluded. Participants who did not respond to Q.3 (haemophilia diagnosis: haemophilia A, haemophilia B, no bleeding disorder) were excluded from the analysis. The participating NGOs distributed the PROBE questionnaires through mail, email, in-person meetings or a combination of methods. The PROBE questionnaire was available in 18 languages with localised language versions in both paper-based and web-based format. A central statistical check for duplicates was run, and three potential duplicates were excluded.

### Ethical approval

Patients' identifier or personal information was not collected as part of the study. Data were collected as anonymous individuals, and study data were transferred and stored at McMaster University.

### PROBE questionnaire

The detail of questionnaire development and feasibility study was described elsewhere.[22] The PROBE questionnaire is organised in four sections, comprising 29 questions. Sections are numbered following the order of presentation in the questionnaire. PROBE PRO domains are covered in Section II. The questions in Section I and III do not cover PRO domains. Only PWH are expected to complete Section III, whereas every participant completes Sections I, II and IV. Section I contains seven questions pertaining to demographic data (country, gender, diagnosis of haemophilia or absence of a bleeding disorder, year of birth, body weight, age

first started and finished school, marital status and children). Section II contains nine questions pertaining to PROs, including general health issues, use of mobility aids or assistive devices, pain (including acute, chronic and pain medications), daily activities, current work or student status, surgeries or procedures and comorbid diseases. Section III contains 12 questions pertaining to clinical aspects of haemophilia (severity of haemophilia, inhibitor status, bleeding history, haemophilia care, treatment regimen, target joints, joint bleeding, range of motion and life-threatening or limb-threatening bleeds). Section IV contains the EuroQol five dimension 5-level instrument (EQ-5D-5L),[24] consisting of questions regarding mobility, self-care, usual activities, pain or discomfort and anxiety or depression, and the EuroQol visual analogue scale (EQ-VAS) of global health[24] was incorporated in the PROBE questionnaire with permission.

## Item scaling and PROBE score calculation

PROs were evaluated only in Section II. The calculation of the PROBE score was based on multiattribute value functions.[25 26] The assessed scores ($X_i$) were converted to returns-to-scale score ($V_i(X_i)$), given that $0 \leq V_i(X_i) < 1$. Q.8 which had a dichotomous response (0=no, 1=yes) produce dichotomous score of 0 and 1. Two questions (Q.10 and Q.15) asked for frequency of the use of pain medication(s) and number of surgeries or invasive procedures. The 6-level and 7-level Likert scales from these two questions were converted to a returns-to-scale score, ranging from 0 to 1. The number of days absent from work or school (Q.14) was converted to returns-to-scale score by dividing by 366. Questions regarding mobility aids, acute pain, chronic pain and comorbid diseases (Q.9, Q.11, Q.12, Q.13 and Q.16) had multiple choices. The scales for these items were calculated based on the cumulative number of choices checked. We apply weight for subitems in each question (if needed). The final score was calculated by summing all of the 11 items scores from the nine questions using additive value function and then scaled so the PROBE Score ranged from 0 to 1 (higher value indicates better health status).

## Data analyses

### Descriptive statistics

Demographic data of study participants were summarised using mean with corresponding SD or median and quartile range as appropriate. Categorical data were summarised using numbers and percentages. Participants who did not respond in Q.3 (disease status; haemophilia A, haemophilia B, haemophilia carrier, other bleeding disorders or no bleeding disorder) were excluded from the analysis. An item distribution analysis to evaluate the proportion of missing data was performed. Floor and ceiling effects were evaluated by the proportion of respondents with scores at floor (minimum score) and ceiling (maximum score), respectively. We predefined that we would have considered a floor or ceiling effect relevant using the

empirical threshold of 15% and a cumulative ceiling or flooring of 50% as proposed by Terwee et al.[27]

## Psychometric analyses

Face and content validity were assessed and reported previously.[22] Test-retest reliability analyses of the PROBE questionnaire were reported elsewhere.[28] In the current study, the following psychometric analyses were carried out.

### Principal axis factor analysis

An exploratory factor analysis of nine questions, pertaining to the PROs (Section II). Principal axis factor analysis with oblique rotation method was performed. The percentage of variance on the items that were explained by the factors was evaluated. Higher percentage indicated strong influence of the factors. The regression coefficients (factor loadings) of the item responses on the retaining factors after factor rotation were calculated.

### Internal consistency reliability

An analysis to confirm the precision of the scale based on the intercorrelations of the items evaluating the same construct was conducted. We hypothesised that the questions asking about pain and the use of medications (Q.10–Q.13) were correlated. Cronbach's alpha was used to determine the correlation between items. Cronbach's alpha coefficient greater than 0.7 was considered to indicate acceptable reliability.[29]

### Convergent validity

The convergent validity of the items in the same construct with the existing, standardised questionnaire was assessed. Specifically, we hypothesised that the items asking about the use of mobility aids and assistive devices correlated with the mobility domain of EQ-5D-5L; the items asking about the use of pain medication, acute and chronic pain (Q.10, Q.11 and Q.12) correlated with pain and discomfort domain of EQ-5D-5L; the items asking about activities of daily living (Q.13) correlated with the self-care and usual activity domains of EQ-5D-5L.[30] Each item of EQ-5D-5L was scored, ranging from level 1 (coded as 1) to level 5 (coded as 5). The health states were converted into a single index value using the UK value set. The correlation between the score from each PROBE item and corresponding EQ-5D-5L domain was calculated. Additionally, the correlation between EQ-5D-5L utility index score and the PROBE Score was assessed. Correlation coefficient (r) was interpreted as: r 0.20–0.39; weak correlation; r 0.40–0.59, moderate correlation; r 0.60–0.79, strong correlation and r 0.80–1.00, very strong correlation.[31]

### Known groups validity

The ability of the PROBE questionnaire to determine the differences between known subgroups was assessed. Participants were classified into groups, according to information collected in Section III, as diagnosis (haemophilia or non-haemophilia), severity of haemophilia (mild, moderate or severe), current inhibitor status (yes

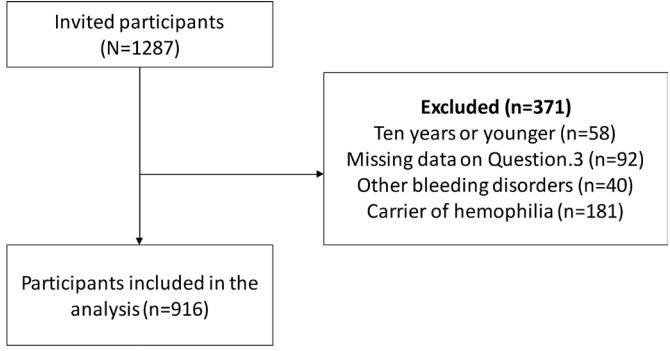

**Figure 1** Flow diagram of participant selection.

or no), number of bleeds in the past year (categorical variable), bleed in the past 2 weeks (yes, no), presence of target joint (yes, no), limitation of range of motion of the joints (yes, no) and life-threatening or limb-threatening bleeding in the past year (yes, no). The PROBE Scores were compared between subgroups using t-test or one-way analysis of variance for the univariate analysis, as appropriate. A priori hypotheses included PWH (as compared with participants without bleeding disorders), patients with severe haemophilia (as compared with mild and moderate haemophilia), patients with current inhibitor (as compared with those without an inhibitor), patients with greater numbers of bleeding, patients who had recent bleeding within the past 2 weeks (as compared with those without), patients with presence of target joint(s) (as compared with those without), patients who had reduced range of motion of any joints (as compared with those without) and patients who had life-threatening or limb-threatening bleeding in the past year (as compared with those without) had worse PROBE scores. The multivariable analysis of the known group validity was conducted using a linear regression. The regression model included age and gender of participants in the analysis. Regression coefficients with corresponding 95% CI were reported. P value less than 0.05 was considered statistically significant.

## RESULTS

### Participants' demographic data

Since inception, NGOs from 21 countries have participated in the PROBE Project. For this study, we performed the analysis using participants' data from the first 17 countries. Figure 1 demonstrates the flow of participant selection for this phase of research. There were 1287 participants who responded to the questionnaire. After excluding haemophilia carriers, other bleeding disorders and missing value (Question 3), and three possible duplicates, the analysis included 916 participants. Demographic data are shown in table 1. Median age of PWHs was lower than that of controls, 33 (quartile 1, quartile 3 of 24, 46) vs 43 (quartile 1, quartile 3 of 34, 54) years. The proportion of male participants in haemophilia group was greater than those in control group (93.7% vs 6.4%). Among patients with haemophilia, most had severe

**Table 1** Participants' characteristics

| Characteristics | Participants (n=916)* |
|---|---|
| Age, median (Q1, Q3) | 37 (27, 48) |
| Diagnosis, n (%) | |
| Haemophilia A | 532 (58.1) |
| Haemophilia B | 82 (8.9) |
| Non-haemophilia | 302 (33.0) |
| Severity of haemophilia†, n (%) | |
| Normal | 3 (0.6) |
| Mild | 54 (10.6) |
| Moderate | 88 (17.3) |
| Severe | 352 (69.3) |
| Do not know | 11 (2.2) |
| Ever been diagnosed with inhibitor†, n (%) | |
| Yes | 70 (14.1) |
| No | 384 (77.2) |
| Do not know | 43 (8.7) |
| Currently have an clinically significant inhibitor, n (%) | 24 (2.6) |
| Sex, n (%) | |
| Male | 685 (74.8) |
| Female | 231 (25.2) |
| Age when started school, median (Q1, Q3) | 6 (5, 6) |
| Year of school or education, median (Q1, Q3) | 15 (12, 18) |
| Married or long-term relationship, n (%) | 581 (69.0) |
| Having children, n (%) | 462 (55.3) |
| Region, n (%) | |
| Africa | 8 (0.9) |
| Western Pacific | 216 (23.6) |
| South America | 343 (37.4) |
| North America | 138 (15.1) |
| Europe | 211 (23.0) |

*After exclusion of three possible duplicates.
†Haemophilia population.
Q1; the first quartile, Q3; the third quartile.

haemophilia. Seventeen participants (2.6%) of PWH had an inhibitor during the study period.

### Descriptive analysis

Table 2 demonstrates item distribution and missing data. Ceiling effect greater than 15% was observed in all but one item (the use of pain medications) in Section II. Similarly, ceiling effect greater than 15% was observed in all domains of EQ-5D-5L. Floor effect greater than 15% was found in four items (problems related to health, bleeding in the past 12 months, limitation of range of motion and life-threatening or limb-threatening bleeding). We observed a higher frequency of ceiling effect among participants without a bleeding disorder as compared

**Table 2** Item distribution and missing data

| Item | Floor (%) | Ceiling (%) | Missing (%) |
|---|---|---|---|
| Patient-reported outcome | | | |
| Q.8 Problem related to health* | 59.1 | 32.3 | 8.6 |
| Q.9 Mobility aids or assistive devices | 0.1 | 0 | 11.5 |
| Q.10 Pain medications | 3.0 | 14.6 | 12.3 |
| Q.11.1 Acute pain (activities) | 0.7 | 33.1 | 12.8 |
| Q.11.2 Acute pain (interference) | 0.3 | 33.2 | 12.8 |
| Q.12.1 Chronic pain (activities) | 1.4 | 32.6 | 13.5 |
| Q.12.2 Chronic pain (interference) | 0.1 | 33.6 | 13.5 |
| Q.13 Daily activities | 0.1 | 42.4 | 14.3 |
| Q.14 Work/school life | 0.1 | 27.8 | 21.8 |
| Q.15 Joint surgery or procedure | 1.3 | 52.4 | 17.0 |
| Q.16 Comorbid diseases | 0 | 56.1 | 0 |
| Haemophilia-related health | | | |
| Q.17 Severity | N/A | N/A | 17.3 |
| Q.18 Inhibitor status | N/A | N/A | 19.1 |
| Q.19 Bleeding in the past 12 months | 16.6 | 8.5 | 18.2 |
| Q.20 Bleeding in the past 2 weeks | N/A | N/A | 18.9 |
| Q.21 Haemophilia treatment centre | N/A | N/A | 19.4 |
| Q.25 Target joints | N/A | N/A | 22.6 |
| Q. 26 Spontaneous joint bleeding | N/A | N/A | 49.4 |
| Q.27 Limitation of range of motion* | 66.6 | 11.4 | 22.0 |
| Q.28 Life-threatening or limb-threatening bleeding* | 15.2 | 62.1 | 22.8 |
| EQ-5D-5L and EQ-VAS | | | |
| Mobility | 1.1 | 32.4 | 21.6 |
| Self-care | 0.7 | 55.0 | 22.3 |
| Usual activities | 0.7 | 37.9 | 22.4 |
| Pain/discomfort | 1.1 | 23.9 | 22.9 |
| Anxiety/depression | 1.6 | 37.3 | 22.8 |
| VAS | 0 | 3.1 | 22.8 |

*Dichotomous outcome.
EQ-5D-5L, EuroQol five dimension 5-level instrument; EQ-VAS, EuroQol visual analogue scale; N/A, not applicable.

with PWH (data not shown). Missing data were 0% to 21.8% in Section II, 18.2% to 49.4% in Section III and 21.6% to 22.9% in Section IV. The median PROBE Score across all participants was 0.78 (mean=0.76, SD=0.16, minimum=0.26 and maximum=0.99).

### Principal axis factor analysis
The principal component factor analysis of the nine questions (11 items) pertaining to the PROs was carried out. These three factors were retained for the following analyses. Table 3 demonstrates factor loadings based on three factors. The items were grouped per factor with their maximum loading.

Factor 1 appears to be the most influential, explaining 87.3% of the variance. There were two items contained in this factor (activities and interference related to chronic pain). Factor 2 contained two items (activities and interference related to acute pain). Factor 3 contained two items pertaining to daily activities and work/school life. All items in the each factor had acceptable factor loadings ($r{\geq}0.3$).[32]

### Internal consistency reliability
The Cronbach's alpha coefficient was acceptable at 0.84.

### Convergent validity
Table 4 shows the correlation coefficients between PROBE items and EQ-5D-5L. The results showed that Q.9 (the use of mobility aids and assistive devices) had a moderate correlation with mobility domain of EQ-5D-5L ($r$=0.42). The pain and discomfort domain of EQ-5D-5L had a moderate to strong correlation with most of the pain related items of the PROBE questionnaire ($r$=0.55 for pain medication, 0.42 for acute pain occurrence, 0.39 for

**Table 3** Principal axis factor analysis, non-orthogonal rotated structure matrix loadings

| Items | Factor 1 | Factor 2 | Factor 3 | Uniqueness |
|---|---|---|---|---|
| Q.8 Problem related to health | 0.1053 | 0.1416 | 0.0277 | 0.7022 |
| Q.9 Mobility aids or assistive devices | −0.1540 | 0.0442 | 0.3470 | 0.7427 |
| Q.10 Pain medications | 0.2065 | 0.0684 | 0.1394 | 0.6174 |
| Q.11.1 Acute pain (activities) | −0.0033 | 0.7963 | 0.0158 | 0.3111 |
| Q.11.2 Acute pain (interference) | 0.0763 | 0.7701 | 0.0005 | 0.2900 |
| Q.12.1 Chronic pain (activities) | 0.8214 | 0.0386 | 0.0329 | 0.2128 |
| Q.12.2 Chronic pain (interference) | 0.8315 | 0.0152 | 0.0092 | 0.1969 |
| Q.13 Daily activities | 0.2573 | 0.0229 | 0.5321 | 0.3854 |
| Q.14 Work/school life | 0.0679 | 0.0477 | 0.5931 | 0.6613 |
| Q.15 Joint surgery or procedure | 0.0489 | 0.0222 | −0.0031 | 0.8356 |
| Q.16 Comorbid diseases | −0.0022 | −0.0832 | 0.0642 | 0.7874 |

acute pain interference, 0.56 for chronic pain occurrence and 0.57 for chronic pain interference). Item related to activities of daily living had a strong correlation with the self-care and usual activities domain ($r=0.65$ and 0.71, respectively). The PROBE score had a strong correlation with the EQ-5D-5L utility index score ($r=0.67$).

### Known groups validity

The regression coefficients of each a priori variable and the PROBE Score were demonstrated in table 5. Participants without a bleeding disorder had a significantly higher PROBE Score when compared with PWH (mean score (SD), 0.87 (0.11) vs 0.71 (0.16), p<0.001). PWH with mild to moderate haemophilia had a slightly higher PROBE Score (mean 0.71, SD 0.16) than severe PWH (mean 0.70, SD 0.16). PWH who had a greater number of bleeding episodes had a significantly lower PROBE Score when compared with those who had less frequent bleeding (p<0.001). Patients who reported bleeding in the past 2 weeks had a significantly lower PROBE score (mean 0.67, SD 0.15) than those without (mean 0.76, SD 0.15). Patients who reported the presence of any target

joints had a significantly lower PROBE score (mean 0.68, SD 0.15) when compared with those who did not (mean 0.78, SD 0.16). Patients who reported three or more spontaneous joint bleeds in the past 6 months had significantly lower PROBE score (mean 0.66, SD 0.14) than those who did not report (mean 0.73, SD 0.14). Patients with reduced range of motion of any joints had a significantly lower PROBE score (mean 0.68, SD 0.14) as compared with those without (mean 0.86, SD 0.15). Patients who previously had life-threatening or limb-threatening bleeding in the past year had a significantly lower PROBE Score (mean 0.62, SD 0.16) when compared with those who did not (mean 0.72, SD 0.15). Table 6 demonstrates multivariable analysis. The findings from multivariable analysis did not change much after adjusting for age and sex.

### DISCUSSION

The psychometric properties of the PROBE questionnaire have been assessed and found that the PROBE

**Table 4** Correlations between PROBE and EQ-5D-5L items (convergent validity)

| EQ-5D-5L | PROBE | Correlation | 95% CI |
|---|---|---|---|
| Mobility | Q.9 Mobility aids | 0.42 | 0.35 to 0.47 |
| Pain and discomfort | Q.10 Pain medications | 0.55 | 0.50 to 0.60 |
| | Q.11.1 Acute pain (activities) | 0.42 | 0.36 to 0.48 |
| | Q.11.2 Acute pain (interference) | 0.39 | 0.32 to 0.45 |
| | Q.12.1 Chronic pain (activities) | 0.56 | 0.51 to 0.61 |
| | Q.12.2 Chronic pain (interference) | 0.57 | 0.52 to 0.62 |
| Self-care | Q.13 Activities of daily living | 0.65 | 0.61 to 0.69 |
| Usual activities | Q.13 Activities of daily living | 0.71 | 0.67 to 0.74 |
| Anxiety | N/A | N/A | N/A |
| Utility index score | Total score | 0.67 | 0.62 to 0.71 |

EQ-5D-5L, EuroQol five dimension 5-level instrument; EQ-VAS, EuroQol visual analogue scale; N/A, not applicable; PROBE, Patient Reported Outcomes, Burdens and Experiences.

**Table 5** Known group validity analyses, univariate analysis

| Subgroup | Total PROBE score, mean (SD) | P values |
|---|---|---|
| **Q.2 Diagnosis** | | |
| Non- haemophilia | 0.87 (0.11) | |
| Haemophilia | 0.71 (0.16) | <0.001 |
| **Q.17 Severity of haemophilia** | | |
| Mild-moderate | 0.71 (0.16) | 0.45 |
| Severe | 0.70 (0.16) | |
| **Q.18 Current inhibitor** | | |
| No | 0.71 (0.19) | 0.35 |
| Yes | 0.67 (0.12) | |
| **Q.19 Number of bleeds in past year** | | |
| 0 bleed | 0.80 (0.14) | <0.001 |
| 1 bleed | 0.85 (0.11) | |
| 2–3 bleeds | 0.75 (0.15) | |
| 4–7 bleeds | 0.74 (0.14) | |
| 8–10 bleeds | 0.70 (0.13) | |
| 11–15 bleeds | 0.68 (0.12) | |
| 16–30 bleeds | 0.65 (0.15) | |
| >30 bleeds | 0.61 (0.15) | |
| **Q.20 Bleed in the past 2 weeks** | | |
| No | 0.76 (0.15) | <0.001 |
| Yes | 0.67 (0.15) | |
| **Q.25 Target joint** | | |
| No | 0.78 (0.16) | <0.001 |
| Yes | 0.68 (0.15) | |
| **Q.26 Spontaneous joint bleeding** | | |
| No | 0.73 (0.15) | 0.0004 |
| Yes | 0.66 (0.14) | |
| **Q.27 Having reduced range of motion** | | |
| No | 0.86 (0.13) | <0.001 |
| Yes | 0.68 (0.14) | |
| **Q.28 Life-threatening bleed** | | |
| No | 0.72 (0.15) | <0.001 |
| Yes | 0.62 (0.16) | |

questionnaire has a strong internal consistency, robust convergent validity and excellent differentiation properties between known groups. We believe these characteristics, jointly with the availability of country specific reference ranges and low impact on NGO resources and time required by the patients make the PROBE questionnaire a tool with great potential for efficient PROs collection in clinical and comparative effectiveness research and for advocacy purposes.

As demonstrated by factor analysis, the core of PROBE revolves around three factors, explaining the majority of the variance in responses. The most influential factor was pain, followed by use of mobility aids or assistive device (complemented by work or school absent days) and comorbidity. No surprise these three elements explain 50% of the variance among different participants: the novelty of PROBE is summarising the assessment of these three domains in a lightweight set of questions for which excellent internal consistency was demonstrated.

The convergent validity analysis showed moderate to strong correlation between PROBE and EQ-5D-5L items, with lower correlations for items concerning pain ($r$ ranged from 0.39 to 0.57), whereas the overall convergence with EQ-5D-5L was confirmed and was intentionally sought to ensure maximising external validity and efficiency for cross-disease comparisons. The pain-related questions in the PROBE questionnaire are related to different aspects (when the pain occurred…, if the pain interfered with any of following…) than EQ-5D-5L.[33] From this perspective, PROBE might be seen as a new hybrid PRO tool, sharing some properties of a generic and some of a disease specific tool. The total PROBE score has a strong correlation with the utility index score of the EQ-5D-5L, both in patients ($r$=0.57) and controls ($r$=0.53), but explores a more specific set of subdomains.

The most important result of this analysis is the demonstration of the discriminative property of the PROBE questionnaire and score. In known group validity analysis, PWH had a significantly lower PROBE Score when compared with the control population (participants without haemophilia). Patients with more frequent bleeds, target joints, reduced range of motion and previous life-threatening or limb-threatening bleeds were demonstrated with a lower PROBE Score (indicating worse health status).

The investigators did not observe a significant difference of the total PROBE Scores among severity of disease as well as current inhibitor status. This outcome may be confounded by bleeding phenotype and joint status. It has been shown that the presence of inhibitor has negative impact on HRQoL in PWH.[34] The regression analysis in this present study revealed that number of bleeds, presence of target joint(s) and limitation of range of motion of any joints, not inhibitor status, were associated with worse health status. There have been studies that reported the negative HRQoL in patients with haemophilia with inhibitor who had poor orthopaedic joint score, who had acute bleeding and who had more frequent bleeding.[35–37] It is important to note that there are relatively a small number of patients with mild-moderate diseases (8.8% and 14.3%, respectively) and those with current inhibitors (4.1%) in this study. The association between inhibitor status and health status of PWH warrant further studies with adequate power.

The PROBE Project has several strengths. First, both PWH and participants without bleeding disorders were recruited, asked PRO questions meaningful to both and derived a PROBE Score applicable to both. Therefore, we were able to compare the health status across health-specific conditions (haemophilia vs non-haemophilia in this study). There is a potential role for the use of the PROBE questionnaire to compare health status between PWH

**Table 6** Coefficients derived from multivariable linear regression analysis

| | Coefficient* | 95% CI | P values |
|---|---|---|---|
| **Q.2 Diagnosis** | | | |
| Non- haemophilia | Control | N/A | N/A |
| Haemophilia | − 0.22 | − 0.25 to − 0.18 | <0.001 |
| **Q.17 Severity of haemophilia** | | | |
| Mild-moderate | Control | N/A | N/A |
| Severe | − 0.003 | − 0.03 to 0.03 | 0.83 |
| **Q.18 Current inhibitor** | | | |
| No | Control | N/A | N/A |
| Yes | − 0.04 | − 0.14 to 0.05 | 0.34 |
| **Q.19 Number of bleeds in past year** | | | |
| 0 bleed | Control | N/A | N/A |
| 1 bleed | 0.04 | − 0.03 to 0.10 | 0.29 |
| 2–3 bleeds | − 0.06 | − 0.11 to 0.001 | 0.06 |
| 4–7 bleeds | − 0.07 | − 0.12 to − 0.01 | 0.02 |
| 8–10 bleeds | − 0.10 | − 0.16 to − 0.03 | 0.002 |
| 11–15 bleeds | − 0.14 | − 0.20 to 0.08 | <0.001 |
| 16–30 bleeds | − 0.15 | − 0.21 to − 0.09 | <0.001 |
| >30 bleeds | − 0.19 | − 0.24 to − 0.13 | <0.001 |
| **Q.20 Bleed in the past 2 weeks** | | | |
| No | Control | N/A | N/A |
| Yes | − 0.09 | − 0.12 to − 0.07 | <0.001 |
| **Q.25 Target joint** | | | |
| No | Control | N/A | N/A |
| Yes | − 0.09 | − 0.13 to − 0.06 | <0.001 |
| **Q.26 Spontaneous joint bleeding** | | | |
| No | Control | N/A | N/A |
| Yes | − 0.09 | − 0.12 to − 0.05 | <0.001 |
| **Q.27 having reduced range of motion** | | | |
| No | Control | N/A | N/A |
| Yes | − 0.14 | − 0.19 to − 0.11 | <0.001 |
| **Q.28 Life threatening bleed** | | | |
| No | Control | N/A | N/A |
| Yes | − 0.10 | − 0.13 to − 0.06 | <0.001 |

*Adjusted from age and sex.
N/A; not applicable.

with any other diseases that share common features, for example, von Willebrand disease, rheumatoid arthritis or osteoarthritis. Second, both school-aged and adult participants were included. The work or school life was assessed in the same manner. As a result, the PROBE questionnaire is valid to implement in participants in all age groups (starting at the not-yet defined age when one is able to comprehend the questionnaire). Third, the questions in the PROBE questionnaire included a standardised observation period in each question stem, generally the past 12 months. This is helpful for participants to respond to each item closest to their actual health condition in a specific time frame.

This PROBE Project also has some limitations, the first being that responsiveness of the PROBE Score has not been validated currently. This study was conducted with a cross-sectional study design. This means participants responded to the questionnaire at a single time. Assessing responsiveness requires a more complicated and demanding study design, which will be addressed in the future. Second, the observation period in the items was up to 12 months, whereas this was chosen

to maximise capturing the impact of rare events, it might introduce recall bias in some participants. Third, a ceiling effect was observed for all except one item concerning PRO as well as all EQ-5D-5L items. The recent study regarding floor and ceiling effects of the EQ-5D-5L in 996 English general population showed that 47.6% of respondents reported the best possible heath state (ceiling effect).[38] In addition, the ceiling effects ranged from 58.4% to 90.8% in the subdomains.[38] The floor effects in the study were relatively lower than the previous reports,[38] probably because sicker participants (PWH) were included.

## CONCLUSION

The psychometric properties of the PROBE questionnaire have been assessed, showing that the PROBE questionnaire has a strong internal consistency, robust convergent validity and excellent differentiation properties between known groups. When compared with EQ-5D-5L, PROBE has a moderate to strong correlation across all domains. The immediate use of the PROBE Score based on these results would be in cross-sectional comparisons among different settings, for example, those defined by different levels of access to care. The PROBE questionnaire has great potential for efficient PROs collection in clinical and comparative effectiveness research and for advocacy purposes. Future applications of PROBE within clinical trials or in longitudinal observational studies will require preliminary demonstration of PROBE test-retest and responsiveness properties, to ensure it is sensitive to meaningful treatment or disease changes over time.

**Author affiliations**
[1]Department of Medicine, McMaster University, Hamilton, Ontario, Canada
[2]Department of Health Research Methods, Evidence, and Impact, McMaster University, Hamilton, Ontario, Canada
[3]Institute for Policy Advancement Ltd, Washington, District of Columbia, USA
[4]Factor VIII Computing, Berkeley, California, USA
[5]Research and Medical Information, National Hemophilia Foundation, New York City, New York, USA
[6]Sol Price School of Public Policy, University of Southern California, Los Angeles, California, USA
[7]Irish Haemophilia Society, Dublin, Ireland
[8]Trinity College Dublin, Dublin, Ireland
[9]Canadian Hemophilia Society, Montreal, Quebec, Canada
[10]Poole College of Management, North Carolina State University, Raleigh, North Carolina, USA
[11]Biostatistics Unit, St Joseph's Healthcare, Hamilton, Ontario, Canada

**Acknowledgements** We thank the haemophilia patient organisations that have participated in the PROBE Project: Fundación de la Hemofilia (Argentina) Cordoba Chapter; Hemophilia Foundation Australia (Australia); Federaçao Brasileira de Hemofilia (Brazil); Canadian Hemophilia Society (Canada); Association Française des Hémophiles (France); Deutsche Hämophiliegesellschaft (Germany); Magyar Hemofilia Egyesulet (Hungary); Irish Haemophilia Society (Ireland); Federazione delle Associazioni Emofilici (Italy); National Hemophilia Network of Japan (Japan); Federación de Hemofilia de la República Mexicana (Mexico); Nederlandse Vereniging van Hemofilie- Patiënten (The Netherlands); Haemophilia Foundation of New Zealand (New Zealand); Haemophilia Foundation of Nigeria (Nigeria); Polish Hemophilia Society (Poland), Federación Española de Hemofilia (Spain); The Haemophilia Society (UK); National Hemophilia Foundation (USA); Asociación Venezolana para la Hemofilia (Venezuela) and Vietnamese Hemophilia Association (Vietnam). We also thank two anonymous reviewers for significant contribution to the quality and clarity of the final version of this manuscript, including some key methodological references.

**Contributors** MS, AI, RC, NF, MBN, DN, BOM, DP and JS conceptualised the study. CC-A and LT performed data collection and statistical analysis. CC-A, AI, MC and MS drafted the manuscript. All authors critically reviewed the manuscript. All authors approved the final manuscript.

**Funding** PROBE is an independent investigator led research project with grant/research support from: Baxalta, now part of Shire; Bayer; Bioverativ, a Sanofi Company; CSL Behring; Novo Nordisk; Roche; and Sobi with collaboration of the US National Hemophilia Foundation.

**Competing interests** Investigators received grants from Baxalta, now part of Shire; Bayer; Bioverativ; CSL Behring, Novo Nordisk; Roche and Sobi and non-financial support from the US National Hemophilia Foundation.

**Patient consent** Not required.

**Ethics approval** Ethical approval was obtained from the Hamilton Integrated Research Ethics Boards. Additional local review ethical board approval was obtained when requested by the local regulation.

**Provenance and peer review** Not commissioned; externally peer reviewed.

**Data sharing statement** The data used in this study are not publicly available due to ethical and legal restrictions.

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
