## [Reviewer comments · BMJ Open]

ARTICLE DETAILS

TITLE (PROVISIONAL)	Psychometric properties of the Patient Reported Outcomes Burdens and Experiences (PROBE) Questionnaire
AUTHORS	Chai-Adisaksopha, Chatree; Skinner, Mark; Curtis, Randall; Frick, Neil; Nichol, Michael; Noone, Declan; O'Mahony, Brian; Page, David; Stonebraker, Jeffrey; Thabane, Lehana; Crowther, Mark; Iorio, Alfonso

VERSION 1 – REVIEW

REVIEWER	Dejan Stevanovic Child psychiatrist, specialist for neuropsychology, Clinic for Neurology and Psychiatry for Children and Youth Belgrade, Serbia
REVIEW RETURNED	04-Feb-2018

GENERAL COMMENTS	I read the study “Psychometric properties of the Patient Reported Outcomes Burdens and Experiences (PROBE) Questionnaire” with great interests and have a few concerns. The authors have to clearly state how the recently published study “Psychometric Properties of the Patient Reported Outcomes Burdens and Experiences (PROBE) Questionnaire“ on http://www.bloodjournal.org/content/130/Suppl_1/5645?sso-checked=true differs from this one and why we need this new study! Abstract Correct the Conclusions to reflect the findings. Such as “Conclusions: The results of this study suggest that PROBE may produce reliable and valid PRO data, in particular that the known-group property of PROBE allow its use in clinical trials health technology assessment studies, routine clinical care or registries. However, additional studies are needed to test responsiveness and sensitivity to change.” Pg 5 Delete the first highlight. it says nothing. Pg 5 Ln 20 correct “win” to “in” Pg 11 “factor analysis” the authors did “Principal component analysis” as a mean of factor analysis, which is incorrect. It has to be done Principal axis factoring. Thus, repeat the analyzes. Pg 14 Ln 50 delete “Psychometric analyses” Pg 15 Ln 34 Delete “The internal consistency reliability was carried out using Cronbach’s alpha. An analysis on pain-related items was performed.” Conclusions Clear conclusions are missing!
--

REVIEWER	Dr Peter Goodwin Manchester Metropolitan University, UK
REVIEW RETURNED	14-Feb-2018

GENERAL COMMENTS	Comments for authors: Thank you for asking me to review this manuscript. I think that this PRO is valuable in its discriminative ability for the population concerned.  • Pg 5, line 20: 'win' should be 'within' this current study. • Pg 9 line 27: How did you prevent numerous responses from the same person? • Data analysis: you don't say how or who analysed the EQ-5D data • It would be useful for the reader to understand which countries were involved. It says on pg 14, line 8, 21 countries have been involved, yet reference 22 (and pg 8 line 6) mentions 17. • Pg 14, line 27:during study period... this is an incomplete sentence. • Please state [with a citation] in the data analysis section why a floor and ceiling effect of 15% is relevant. • Did you find different floor and ceiling effects between the PWH and the non PWH? Is this relevant? Would it have explained the differences between groups? • Please state how you dealt with missing data, if at all. • Table 4. Please standardise the confidence intervals. • Were there variations between nationality? • Pg 18, line 50: place insert a space between 'from' and '21'. • Pg 18, line 55: I think the term 'regardless of languages and cultures' is incorrect. You have grouped all languages and cultures together. It is not that you have not regarded them.
---

VERSION 1 – AUTHOR RESPONSE

Reviewer: 1

Reviewer Name

Dejan Stevanovic

Institution and Country

Child psychiatrist, specialist for neuropsychology

Clinic for Neurology and Psychiatry for Children and Youth Belgrade, Serbia

Please state any competing interests or state 'None declared':

None declared.

Please leave your comments for the authors below

I read the study "Psychometric properties of the Patient Reported Outcomes Burdens and Experiences (PROBE) Questionnaire" with great interests and have a few concerns.

The authors have to clearly state how the recently published study "Psychometric Properties of the Patient Reported Outcomes Burdens and Experiences (PROBE) Questionnaire" on http://www.bloodjournal.org/content/130/Suppl_1/5645 differs from this one and why we need this new study!

The referenced publication was a preliminary report of the present study, submitted as an abstract to the 2017 ASH Annual Meeting. The abstract was based on a smaller sample size of 677 participants as compared to 916 in this manuscript. Based on the feed-back received on the abstract presentation, we have refined our analysis and re-run it on the final dataset. This manuscript provides substantially more detail on methods and results that were not previously reported in the ASH meeting proceedings, and is the single full manuscript we intend to publish. We believe it provides unique and important foundation in the literature for future use of PROBE.

Abstract

Correct the Conclusions to reflect the findings. Such as "Conclusions: The results of this study suggest that PROBE may produce reliable and valid PRO data, in particular that the known-group property of PROBE allow its use in clinical trials health technology assessment studies, routine clinical care or registries. Additional studies are needed to test responsiveness and sensitivity to change."

We thank the reviewer for this suggestion. We revised the conclusion in the abstract and the main

body of the paper accordingly.

Pg 5 Delete the first highlight. it says nothing.

We deleted the first highlight and added additional clarifying text.

Pg 5 Ln 20 correct "win" to "in"

We thank the reviewer. We have corrected the typographical error.

Pg 11 "factor analysis" the authors did "Principal component analysis" as a mean of factor analysis, which is incorrect. It has to be done Principal axis factoring. Thus, repeat the analyzes.

We have repeated the analysis using Principal axis factor method. The main text and table were also revised accordingly.

Pg 14 In 50 delete "Psychometric analyses"

We deleted the phrase as suggested.

Pg 15 In 34 Delete "The internal consistency reliability was carried out using Cronbach's alpha. An analysis on pain-related items was performed."

The sentences were deleted as suggested.

Conclusions

Clear conclusions are missing!

We have revised the conclusion of the paper to more comprehensively report the findings and limitations of the study. Here is how the final paragraph reads now:

The psychometric properties of the PROBE questionnaire have been assessed, showing that the PROBE questionnaire has a strong internal consistency, robust convergent validity and excellent differentiation properties between known groups. When compared to EQ-5D-5L, PROBE has a moderate to strong correlation across all domains. The immediate use of the PROBE score based on these results would be in cross-sectional comparisons among different settings, e.g. those defined by different levels of access to care. The PROBE questionnaire has great potential for efficient PROs collection in clinical and comparative effectiveness research, and for advocacy purposes. Future applications of PROBE within clinical trials or in longitudinal observational studies will require preliminary demonstration of PROBE test-retest and responsiveness properties, to ensure it is sensitive to meaningful treatment or disease changes over time.

Reviewer: 2

Reviewer Name

Dr Peter Goodwin

Institution and Country

Manchester Metropolitan University, UK

Please state any competing interests or state 'None declared':

None declared

Please leave your comments for the authors below

Comments for authors:

Thank you for asking me to review this manuscript.

I think that this PRO is valuable in its discriminative ability for the population concerned.

- Pg 5, line 20: 'win' should be 'within' this current study.

We have revised the sentence.

- Pg 9 line 27: How did you prevent numerous responses from the same person?

Thank you for the opportunity to clarify. Participants were instructed to respond the questionnaire (either paper-based or web-based) a single time. A central statistical check for duplicates based on age, country, number of bleeds and number of target joints identified only 3 possible duplicates in the entire database. Page 9, line 7 and 15 were modified/added to mention the concept. Page 15, line 7, and table 1 have been modified accordingly.

- Data analysis: you don't say how or who analysed the EQ-5D data

We added text on page 12 to clarify the methods regarding analysis of the EQ-5D-5L data.

"Each item of EQ-5D-5L was scored, ranging from level 1 (coded as 1) to level 5 (coded as 5). The health states were converted into a single index value utilizing the United Kingdom value set. The correlation between the score from each PROBE item and EQ-5D-5L was calculated."

- It would be useful for the reader to understand which countries were involved. It says on pg 14, line 8, 21 countries have been involved, yet reference 22 (and pg 8 line 6) mentions 17.

We apologize for this confusion. Overall, we have collected data from 21 countries since inception of the study. However, for this manuscript, we performed the analysis using participants' data from the first 17 countries. We added a sentence on page 14 to clarify the number of countries included in the dataset for our analysis.

“Since inception, NGOs from 21 countries have participated in the PROBE project. For this study, we performed the analysis using participants' data from the first 17 countries.”

The reasons for individual participant exclusion within the 17 countries in this phase of the study are provided in figure 1.

- Pg 14, line 27:during study period... this is an incomplete sentence.

We revised the sentence on page 14.

“Seventeen participants (2.6%) of PWH had an inhibitor during the study period.”

- Please state [with a citation] in the data analysis section why a floor and ceiling effect of 15% is relevant.

Thank you for the opportunity to explain more clearly our assumption. We used the definition proposed in ref 26. Essentially, the authors propose the following: “Floor or ceiling effects are considered to be present if more than 15% of respondents achieved the lowest or highest possible score, respectively [41]. [...] We give a positive rating for (the absence of) floor and ceiling effects if no floor or ceiling effects are present in a sample size of at least 50 patients.” They cite in turn [41] McHorney CA, Tarlov AR. Individual-patient monitoring in clinical practice: are available health status surveys adequate? *Qual Life Res* 1995;4:293e307. We added the following sentence on page 11: % “We pre-defined we would have considered a floor or ceiling effect relevant using the empirical threshold of 15% and a cumulative ceiling or flooring of 50% as proposed by Terwee et al 26.”

- Did you find different floor and ceiling effects between the PWH and the non PWH? Is this relevant? Would it have explained the differences between groups?

We thank the reviewer for this comment. We found that participants without hemophilia had higher frequency of ceiling effect (good status) as compared to PWH (data not shown). This would be explained by the health status of participants without hemophilia was better than PWH's. The sentence was added on page 15.

“We observed higher frequency of ceiling effect among participants without bleeding disorder as compared to those with hemophilia (data not shown).”

- Please state how you dealt with missing data, if at all.

As indicated in Figure 1, participants with missing data on Question 3 (diagnosis) were excluded from the dataset, as they data would have been uninterpretable. For the remaining 916 participants, we assumed other missing data were random. A complete case analysis was performed. Percent missing by item was reported in table 2.

- Table 4. Please standardise the confidence intervals.

Table 4 was revised as suggested.

- Were there variations between nationality?

We have conducted a separate study and analysis on a larger dataset to explore regional variations in the cross-cultural, international implementation of the PROBE study. It is our intention to report this analysis in a subsequent dedicated manuscript, as we feel this is already complicated enough. In brief, after adjusting for sex, hemophilia diagnosis and age, nationality (grouped by geographical regions) was not significantly associated with the change of the PROBE score.

- Pg 18, line 50: place insert a space between 'from' and '21'.

We revised the sentence on page 18.

- Pg 18, line 55: I think the term 'regardless of languages and cultures' is incorrect. You have grouped all languages and cultures together. It is not that you have not regarded them.

We agree with the reviewer and have revised the sentence on page 18 accordingly. This will be indeed the focus of the future manuscript described in answering your previous question. We have taken the concept out of the list of points of strength.

VERSION 2 – REVIEW

REVIEWER	Dr Peter Goodwin Manchester Metropolitan University, UK
REVIEW RETURNED	27-Apr-2018
GENERAL COMMENTS	Thank you for your amendments.